# Application of CPI cutoff value based on parentage testing of duos and trios typed by four autosomal kits

**Hongmei Gao**[1,2], **Chang Wang**[1,2], **Ruxia Zhang**[3], **Hanyang Wu**[4], **Shanhui Sun**[1,2], **Dongjie Xiao**[1,2], **Yunshan Wang**[1,2], **Maoxiu Zhang** [1,2]*

**1** Jinan Central Hospital Affiliated to Shandong University, Jinan, Shandong, China, **2** Jinan Di'en Forensic Institute of Jinan Central Hospital, Jinan, Shandong, China, **3** Department of Nursing, Shandong Medical College, Jinan, Shandong, China, **4** Cheeloo College of Medicine, Shandong University, Jinan, Shandong, China

* zhangmaoxiujn@126.com

**Data Availability Statement:** All relevant data are within the manuscript and its Supporting Information files.

## Abstract

In this study, we analyzed the application of four autosomal kits and the sensitivity of the combined paternity index (CPI) cutoff value (CPI$\geq$10000) in parentage testing. First, 1442 real trios and 803 real duos were tested using the Goldeneye 25A kit. The Goldeneye 25A kit covers the autosomal short tandem repeat (STR) loci of the other three kits, so we calculated the CPI value of every case for the four kits. Second, three complex close relative kinship cases were also analyzed to evaluate the application of the CPI cutoff value. The CPI values of all trio cases were higher than 10000 using the four kits; the CPI values of all duo cases were higher than 10000 using the Goldeneye 25A kit; and the CPI values of a portion of the duo cases were lower than 10000 using the other three kits. In the three complex close relative cases, the alleged father or mother was not excluded using 40 autosomal STRs. Adding X chromosome short tandem repeats (X-STR) and samples of biological fathers or mothers, the conclusions were confirmed. The four kits were adequate to draw conclusions in the trio cases; the Goldeneye 25A Kit was adequate to draw conclusions in the duo cases; and the other three kits were not sufficient for a portion of the duo cases. The CPI cutoff value was sensitive for real trio and duo cases. In complex close relative kinship cases, high CPI values may result in false conclusions.

## Introduction

In forensic laboratories, autosomal short tandem repeats (STRs) are used in forensic parentage testing [1]. At present, the commercial kits include 13 Combined DNA Index System (CODIS) STRs. The number of STRs in the kits determines the system efficiency. The combined power of exclusion and combined power of discrimination of kits should be above 0.9999 for duo and trio cases. For duo cases, the STR number was more than that of trio cases due to the lack of a biological mother or father [2]. STR locus mutation reduces the combined paternity index (CPI) value of the case and may lead to inconclusive results. It is necessary to increase the

**Funding:** The authors received no specific funding for this work.

**Competing interests:** The authors have declared that no competing interests exist.

number of STR loci to obtain an adequate CPI value for the cases. High CPI values could also result in false conclusions when the STR gene type of two unrelated people matched [3]. X chromosome short tandem repeats (X-STR) and Y chromosome short tandem repeats (Y-STR) are inherited in a unique, non-Mendelian fashion. Y-STRs are passed down from father to son. A father's X-STRs are passed down only to his daughter, and a mother's X-STRs are inherited by her sons and daughters. X-STR and Y-STR analyses are important for supplementing autosomal STR kit results in forensic cases [4–6].

In Chinese forensic laboratories, the CPI cutoff value of autosomal STRs for parentage testing is provided. In duo and trio cases, the alleged father or mother is confirmed when the CPI is ≥10000. The results are inconclusive when the CPI is 0.0001<CPI<10000, and the alleged father or mother is excluded when the CPI is<0.0001 [7]. The Goldeneye 25A kit includes 23 autosomal STRs. It covers all the STRs of Goldeneye 20A, AmpFlSTR SinoFiler and AmpFlSTR Identifiler kits. In the present study, we evaluated the application of 4 autosomal STR kits and the sensitivity of the CPI cutoff value (CPI ≥10000) based on routine parentage testing of duos and trios in our laboratory. At the same time, 3 complex close relative cases were also analyzed, which required the addition of other autosomal STRs and X-STRs to confirm the conclusions.

## Materials and methods

### Samples

In this study, 1442 real trio parentage testing cases, 803 real duo cases (including 412 motherless and 391 fatherless cases) and 3 complex close relative kinship cases were analyzed to evaluate the use of the CPI cutoff value. All the cases were typed using the Goldeneye 25A Kit in our laboratory. Three complex relative kinship cases were also tested, adding up to more autosomal SRTs and X-STRs for real conclusions. For all the cases, the parents or guardians of the children signed the informed consent forms with our laboratory. At the same time, the study was approved by the Ethics Committee of Jinan Central Hospital affiliated with Shandong University, China.

### STR loci of each kit

1. Goldeneye 25A kit includes 23 STRs (D2S441, TPOX, D22S1045, D7S820, D1S1656, Penta E, D10S1248, D8S1179, D5S818, D19S433, D16S539, CSF1PO, Penta D, D3S1358, vWA, D2S1338, D18S51, D6S1043, D13S317, TH01, D12S391, D21S11, FGA);

2. Goldeneye 20A kit includes 19 STRs (TPOX, D7S820, Penta E, D8S1179, D5S818, D19S433, D16S539, CSF1PO, Penta D, D3S1358, vWA, D2S1338, D18S51, D6S1043, D13S317, TH01, D12S391, D21S11, FGA);

3. AmpFlSTR Identifiler kit includes 15 STRs (TPOX, D7S820, D8S1179, D5S818, D19S433, D16S539, CSF1PO, D3S1358, vWA, D2S1338, D18S51, D13S317, TH01, D21S11, FGA);

4. AmpFlSTR SinoFiler kit includes 15 STRs (D7S820, D8S1179, D5S818, D19S433, D16S539, CSF1PO, D3S1358, vWA, D2S1338, D18S51, D6S1043, D13S317, D12S391, D21S11, FGA);

5. Goldeneye 22NC kit includes 4 STRs found in the Goldeneye 25A Kit (D3S1358, D2S441, D1S1656, D10S1248) and 17 other autosomal STRs (D4S2366, D6S477, GATA198B05, D15S659, D8S1132, D3S3045, D14S608, D17S1290, D3S1744, D18S535, D13S325, D7S1517, D10S1435, D11S2368, D19S253, D7S3048, D5S2500);

6. Goldeneye 17X kit includes 16 X-STRs (DXS6795, DXS9902, DXS8378, HPRTB, GATA165B12, DXS7132, DXS7424, DXS6807, DXS6803, GATA172D05, DXS6800, DXS10134, GATA31E08, DXS10159, DXS6789, DXS6810).

## STR typing

DNA was extracted using Chelex-100 [8]. Goldeneye 25A, Goldeneye 22NC and Goldeneye 17X kits were used for PCR according to the manufacturer's instructions. Autosomal STR and X-STR were typed by the ABI PRISM 3500 Genetic Analyzer. The data were analyzed using GeneMapper ID-X 1.3 software.

## Statistical analysis

The CPI values of duo and trio cases for four kits were calculated according to the Specifications of Parentage Testing in China [7]; gene frequencies of 40 autosomal STRs from the Shandong Han population in China were used for CPI values of all cases [9, 10]. The CPI values of different groups (including trio, duo cases, and different kits) were compared using the Mann-Whitney U test; the sensitivity of the CPI cutoff value was compared using the chi-square test. The results were considered significant if $P < 0.05$.

## Results

### Application of CPI cutoff value for parentage testing of trios

As shown in Table 1, the sensitivity of the CPI cutoff value (CPI≥10000) was high for trio cases. The CPI values of all the trio cases were higher than 10000 typed by the four autosomal kits, and every case had a confirmed conclusion. CPI values from high to low were the Goldeneye 25A kit> Goldeneye 20A kit>AmpFlSTR SinoFiler kit>AmpFlSTR Identifiler kit. There were significant differences between the four kits ($P < 0.05$).

### Application of CPI cutoff value for parentage testing of duos

As shown in Table 1, for every trio case, we regarded it as two real duo cases: a motherless case and a fatherless case. We found that the CPI values of all duo cases were higher than 10000 when typed by the Goldeneye 25A kit. The CPI value of a portion of the duo cases was 0.0001<CPI<10000 typed by the other three kits, so the cases had inconclusive results. The rate of inconclusive results from low to high was in the order Goldeneye 20A kit<AmpFlSTR SinoFiler kit<AmpFlSTR Identifiler kit. The CPI values and rate of inconclusive results were

**Table 1. Sensitivity of the CPI cutoff value for real trio and duo cases.**

| Cases | Scope | 25A (23 STRs) | | 20A (19 STRs) | | Sino (15 STRs) | | ID (15 STRs) | |
|---|---|---|---|---|---|---|---|---|---|
| | | **A** | **B** | **A** | **B** | **A** | **B** | **A** | **B** |
| Trios | AF-C-M(n = 1442) | 1442 | 0 | 1442 | 0 | 1442 | 0 | 1442 | 0 |
| | AF-C (n = 1442) | 1442 | 0 | 1416 | 26 | 1269 | 173 | 1101 | 341 |
| Duos | AM-C (n = 1442) | 1442 | 0 | 1425 | 17 | 1288 | 154 | 1142 | 300 |
| | AF-C (n = 412) | 412 | 0 | 407 | 5 | 363 | 49 | 322 | 90 |
| | AM-C (n = 391) | 391 | 0 | 389 | 2 | 339 | 52 | 296 | 95 |

25A: Goldeneye 25A Kit; 20A: Goldeneye 20A Kit; ID: AmpFlSTR Identifiler Kit;

Sino: AmpFlSTR SinoFiler Kit. AF: alleged father; AM: alleged mother; M: mother;

C: child. A: CPI≥10000; B: 0.0001<CPI<10000.

**Table 2. CPI values of three complex close relative kinship cases.**

| Cases | Scope | Tested by 25A | CPI Values | Adding 22NC test | CPI Values |
|---|---|---|---|---|---|
| 1 | AF-C-M | D6S1043,D22S1045 | $3.6787 \times 10^2$ | GATA198B05,D15S659, D14S608 | 0.8825 |
| | AF-C | D6S1043 | $2.03447 \times 10^3$ | D15S659, D14S608 | $3.5647 \times 10^3$ |
| 2 | F-C-AM | TH01, D21S11 | $5.2048 \times 10^6$ | D8S1132, D17S1290, D18S535 | $2.3643 \times 10^4$ |
| | AM-C | 0 | $6.4469 \times 10^6$ | 0 | $1.1392 \times 10^{13}$ |
| 3 | F-C-AM | D8S1179,D19S433 | $2.7814 \times 10^7$ | D3S3045,D14S608, D17S1290,D3S1744, D11S2368, D19S253 | 0.00043 |
| | AM-C | D8S1179, D19S433 | $2.9800 \times 10^5$ | D14S608, D3S1744, D11S2368, D19S253 | 0.00012 |

F = father; M = mother; AF = alleged father; AM = alleged mother; C = child.

significantly different between the three kits ($P<0.05$). For the 412 motherless and 391 father-less cases, the results were consistent with those of the duo cases separated from the trio cases. The CPI value and sensitivity of the CPI cutoff value were not significantly different between the motherless and fatherless cases ($P<0.05$).

### Three complex close relative kinship trio cases

As shown in Table 2, in the first trio case, the child was a boy and the alleged father was the child's uncle. Typed by the Goldeneye 25A kit, only 2 STR loci had no alleles from his uncle, with a CPI = $3.6787 \times 10^2$. After adding up to 40 STR loci using the Goldeneye 22NC kit, 5 STR loci had no alleles from his uncle, with a CPI = 0.8825. Without the mother's information, as a duo case, only 3 STRs of the child had no alleles from his uncle, with a CPI = $3.5647 \times 10^3$. We could not exclude the possibility that his uncle was the biological father. The Y-STR kit was not used for the child and the child's uncle, because the results of the Y-STR gene type were same for them. To obtain reliable conclusions, we added the STRs gene type of his biological father for comparison. The alleles at 40 STRs of the biological father matched that of the child (data not shown). The alleged fathers were excluded based on the results. In the second case, the child was a girl. The alleged mother was her aunt, who was typed using the Goldeneye 25A kit. As a trio case, only two STR loci had no alleles from her aunt, with a CPI = $5.2048 \times 10^6$. Adding up to 40 STRs, 5 STRs had no alleles from her aunt, with a CPI = $2.3643 \times 10^4$. As a duo case, all 40 STRs of child had alleles from her aunt, with a CPI = $1.1392 \times 10^{13}$. We added the X-STR kit test, and only one X-STR (DXS10134) had no alleles from her aunt (Table 3). Adding a sample from her biological mother for comparison, the child's alleles at 40 autosomal STRs and 16 X-STRs came from her biological mother (data not shown). The alleged mother

**Table 3. X-STR gene type results of case 2.**

| X-STR | F | C | AM | X-STR | F | C | AM |
|---|---|---|---|---|---|---|---|
| DXS6795 | 13 | 11, 13 | 11, 11 | DXS6803 | 11 | 11, 12 | 11.3, 12 |
| DXS9902 | 10 | 10, 11 | 11, 12 | GATA172D05 | 9 | 9, 10 | 6, 10 |
| DXS8378 | 10 | 10, 11 | 10, 11 | DXS6800 | 16 | 16, 21 | 16, 21 |
| HPRTB | 14 | 14, 14 | 12, 14 | DXS10134 | 36 | 36, 37 | 33, 38 |
| GATA165B12 | 10 | 10, 11 | 10, 11 | GATA31E08 | 11 | 11, 12 | 11, 12 |
| DXS7132 | 15 | 12, 15 | 12, 14 | DXS10159 | 26 | 24, 26 | 24, 27 |
| DXS7424 | 16 | 15, 16 | 15, 16 | DXS6789 | 21 | 15, 21 | 15, 20 |
| DXS6807 | 15 | 15, 15 | 14, 15 | DXS6810 | 19 | 19, 19 | 18, 19 |

F = father; C = child; AM = alleged mother.

**Table 4. X-STR gene type results of case 3.**

| X-STR | AM | C | X-STR | AM | C |
|---|---|---|---|---|---|
| DXS6795 | 11, 13 | 11 | DXS6803 | 11, 12 | 12.3 |
| DXS9902 | 10, 12 | 12 | GATA172D05 | 8, 11 | 11 |
| DXS8378 | 10, 12 | 12 | DXS6800 | 16, 16 | 16 |
| HPRTB | 12, 13 | 12 | DXS10134 | 34, 37 | 37 |
| GATA165B12 | 9, 11 | 11 | GATA31E08 | 10, 10 | 10 |
| DXS7132 | 14, 16 | 14 | DXS10159 | 24, 25 | 24 |
| DXS7424 | 16, 16 | 13 | DXS6789 | 16, 16 | 20 |
| DXS6807 | 11, 11 | 11 | DXS6810 | 18, 19 | 18 |

AM = alleged mother; C = child.

was excluded based on the results. In the third case, we did not exclude the possibility that his aunt was his biological mother based on 40 autosomal STR loci. The client could not provide a sample of the biological mother. The child was a boy, so we added the X-STRs test. We found that the alleles at three X-STR loci of the child did not come from his aunt. Therefore, we ruled out the possibility that his aunt was his biological mother (Tables 2 and 4).

## Discussion

Autosomal STR analysis is the primary method for parentage testing in forensic laboratories. The STR gene type of a child is based on his father and mother. In a real trio case, the paternity index (PI) of every STR locus can be calculated according to the STR gene type of the child's mother and alleged father. In a real duo case, the PI of every STR locus can be calculated between two samples that share at least one allele. For one case, the CPI of all kit STR loci can be determined by multiplying the PI of every locus. The results of parentage testing are directly decided by CPI values. The conclusion of the probability of paternity can be determined from the CPI cutoff value set by the Specification of Parentage Testing in China [7].

In the present study, we calculated the CPI values of 1442 real trio cases for 4 kits, including the Goldeneye 25A, Goldeneye 20A, AmpFlSTR SinoFiler and AmpFlSTR Identifiler kit. All the CPI values of the trios were above 10000. The alleged father was the biological father without an STR mutation, and the CPI value was significantly different among the four kits ($P<0.05$). The four kits were sufficient to draw conclusions in the trio cases. In all the duo cases, including motherless and fatherless cases, typed by the Goldeneye 25A kit, all the CPI values of the duos were higher than 10000, but some of the CPI values were $0.0001<CPI<10000$ when tested by the other three kits. The system efficiency of the four kits from high to low were the Goldeneye 25A kit>Goldeneye 20A kit>AmpFlSTR SinoFiler kit>AmpFlSTR Identifiler kit. The AmpFlSTR SinoFiler Kit is based on the AmpFlSTR Identifiler kit. In the AmpFlSTR SinoFiler kit, D6S1043 and D12S39 with a high power of exclusion substitute for the THO1 and TPOX loci, so the system efficiency of the AmpFlSTR SinoFiler kit is higher than the AmpFlSTR Identifiler kit. This result was consistent with another study [11]. In the present study, the results showed that the four kits could meet the requirements of routine parentage testing of trios. The Goldeneye 25A kit was sufficient to draw conclusions in the duo cases, but the other three kits were insufficient for a portion of the duo cases. In order to resolve the duo cases with a low CPI value, we should add more autosomal STR loci to increase the CPI value.

For complex close relative cases, the alleged father usually was the brother of the biological father; the alleged mother was the mother's sister. In the first case, the alleged father was the

child's uncle, and the CPI of the trio was 0.0001<CPI<10000 using 23 or 40 STR loci (Table 2). Y-STR analysis was useless, because they come from the same paternal line. To obtain a certain conclusion, we had to add 40 autosomal STR gene types of his biological father for comparison (data not shown). In the second case, the alleged mother was her aunt. The CPI value of the trio was higher than 10000, with 5 STR loci having no alleles from her aunt. Without a biological father, as an alleged mother and child case, the child's alleles at 40 STRs came from the alleged mother, and the CPI value was $1.1392 \times 10^{13}$. Alleles at only one X-STR locus did not come from the alleged mother (Tables 2 and 3). Considering that alleles at 40 autosomal STRs and 16 X-STRs came from her biological mother (data not shown), we excluded the possibility that her aunt was his biological mother. In the third case, we did not exclude that his aunt was his mother when using 23 or 40 autosomal STRs, but we excluded his aunt using the X-STR gene type (Tables 2 and 4). In the second and third cases, two STR mutations were found after testing by the Goldeneye 25A kit, and the CPI>10000. If we did not know the expected results of the identification, it would result in a false conclusion that the alleged mother was the child's biological mother with loci mutation. Therefore, evaluating simple CPI values of duo or trio cases may lead to false conclusions [2, 12]. Mutations of the STR loci are relatively common in forensic cases. They reduce the CPI value of the case, and affect the conclusion of parentage testing with a low CPI value. More autosomal STRs should be added to confirm the mutation [13–15]. One or two STR loci mismatches may be due to mutational events. Sometimes, we should consider the mutational events as error events. Considering the possibility that autosomal STR loci mutations may occur, it is necessary to increase the number of the required STR loci and supplement the samples of the triplet. In this way, the identification errors could be greatly decreased. In complex close relative cases, there is great similarity of STR gene type among the close relatives, because the alleged father or mother is in the child's immediate family. In some cases, all the limited STR loci gene types of the child matched that of the alleged father or mother. In fact, they were not the child's biological father or mother. Sometimes, high CPI values can lead to false conclusions [16]. In complex close relative cases, the genetic background of the case should be considered, and more autosomal STRs, X-STRs and Y-STRs should be used for further confirmation.

## Conclusion

In summary, the four autosomal kits were adequate to draw conclusions in the trio cases, and the Goldeneye 25A kit was adequate to draw conclusions in the duo cases. The other three autosomal kits were not sufficient for satisfactory conclusions for all duo cases. For complex close relative kinship cases, more autosomal STR loci and other genetic markers are necessary. The CPI cutoff value (CPI≥10000) is satisfactory for all trio cases and most duo cases. In some complex close relative cases, high CPI values may result in false conclusions.

## Supporting information

**S1 Data.**
(XLS)

## Acknowledgments

We thank Professor Cheng-tao Li (Academy of Forensic Science, Ministry of Justice, Shanghai, China) for data analysis advice that was instrumental for this study.

## Author Contributions

**Data curation:** Hongmei Gao, Chang Wang, Ruxia Zhang, Hanyang Wu.

**Formal analysis:** Ruxia Zhang, Maoxiu Zhang.

**Project administration:** Yunshan Wang.

**Resources:** Shanhui Sun, Dongjie Xiao.

**Software:** Hongmei Gao, Chang Wang.

**Writing – original draft:** Maoxiu Zhang.

**Writing – review & editing:** Maoxiu Zhang.

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
