## [Decision Letter · Decision Letter 0]

9 Oct 2019

PONE-D-19-24597

The Title: Application of CPI Cutoff Value Based on Parentage Testing of Duos and Trios Typed by Four Autosomal Kits

PLOS ONE

Dear Dr Zhang,

Thank you for submitting your manuscript to PLOS ONE. After careful consideration, we feel that it has merit but does not fully meet PLOS ONE’s publication criteria as it currently stands. Therefore, we invite you to submit a revised version of the manuscript that addresses the points raised during the review process.

We would appreciate receiving your revised manuscript by Nov 23 2019 11:59PM. To enhance the reproducibility of your results, we recommend that if applicable you deposit your laboratory protocols in protocols.io, where a protocol can be assigned its own identifier (DOI) such that it can be cited independently in the future. For instructions see: http://journals.plos.org/plosone/s/submission-guidelines#loc-laboratory-protocols

We look forward to receiving your revised manuscript.

Kind regards,

Alexandru Rogobete, Ph.D., M.Sc., Clin Res

Academic Editor

PLOS ONE

Journal Requirements:

3. In your data availability statement you write, "All relevant data are within the paper and its Supporting Information files." Please ensure you have provided the individual data points used to create the figures and determine means, medians and variance measures presented in the results, tables and figures (http://journals.plos.org/plosone/s/data-availability#loc-faqs-for-data-policy). If these data cannot be publicly deposited or included in the supporting information, e.g. due to patient privacy or ownership by a third party, explain why and explain how researchers may access them.

'The funders had no role in study design, data collection and analysis, decision to publish, or preparation of the manuscript.'

Please provide an amended Funding Statement that declares *all* the funding or sources of support received during this specific study (whether external or internal to your organization) as detailed online in our guide for authors at http://journals.plos.org/plosone/s/submit-now.  

Please state what role the funders took in the study.  If any authors received a salary from any of your funders, please state which authors and which funder. If the funders had no role, please state: "The funders had no role in study design, data collection and analysis, decision to publish, or preparation of the manuscript."

Additional Editor Comments (if provided):

Reviewers' comments:

Reviewer's Responses to Questions

**Comments to the Author**

1. Is the manuscript technically sound, and do the data support the conclusions?

Reviewer #1: Yes

Reviewer #2: Partly

2. Has the statistical analysis been performed appropriately and rigorously? 

Reviewer #1: I Don't Know

Reviewer #2: I Don't Know

3. Have the authors made all data underlying the findings in their manuscript fully available?

Reviewer #1: Yes

Reviewer #2: Yes

4. Is the manuscript presented in an intelligible fashion and written in standard English?

Reviewer #1: Yes

Reviewer #2: No

5. Review Comments to the Author

Reviewer #1: The manuscript entitled "Application of CPI Cutoff Value Based on Parentage Testing of Duos and Trios Typed by Four Autosomal Kits" by Hong-mei Gao provides quality results.

I also described some parts that could be improved where the sentences and paragraphs are even more difficult to follow.

In the paragraph "STR loci mutations affect the conclusions of parentage testing with a low CPI value, and more autosomal

STRs should be added to confirm the mutation" can you explain it's not clear written.

When you use abbreviations, for the first time in your manuscript, you have to give details about that.

I recommend that a native speaker of English review the manuscript to improve word choice, sentence structure, and grammar.

The conclusion need to clear and specific, with 3 short conclusion.

Thanks for the opportunity to read the manuscript.

Reviewer #2: The language and structure of the article are so unclear that the merit can't be assessed. Most of the sentences are overly long, complicated, as well as ungrammatical in some instances. Throughout the article the language is neither clear, nor concise. Typographical and spelling errors have also been encountered.

The article gives relations to three cases that are not included in the study groups described in the methods. Therefore part of the article rather looks as a case report (in this case sample sizes are not large enough to produce robust results).

Regarding the tables of this manuscript, they are not all placed directly after the paragrapgh where they were first cited (see Table 2).

The References formatting does not respect the style of the journal (does not list the first six authors followed by et.al.). Some of the title used as references are rather outdated (over 20 years old).

Laboratory protocols were not made fully availabe. Detailed protocols could enhance the reproductibility of the results.

In part the Discussions only repeat the results without interpreting them.

6. PLOS authors have the option to publish the peer review history of their article (what does this mean?). If published, this will include your full peer review and any attached files.

Reviewer #1: No

Reviewer #2: No

---

## [Author Response · Author response to Decision Letter 0]

25 Oct 2019

We are sorry for some flaws and incorrections appeared in this paper and appreciate the hard-work of your technical staff for revising. Here are the answers for the comments raised by the reviewers in the paper and we made changes in the paper accordingly.

Reviewer #1:

1. In the paragraph "STR loci mutations affect the conclusions of parentage testing with a low CPI value, and more autosomal STRs should be added to confirm the mutation" can you explain it's not clear written.

Answer: We explained it in discussion. “One or two STR loci mismatches may be due to mutational events. Sometimes, we should consider the mutational events as error events. Considering the possibility that autosomal STR loci mutations may occur, it is necessary to increase the number of the required STR loci and supplement the samples of the triplet. In this way, the identification errors could be greatly decreased.”

2. When you use abbreviations, for the first time in your manuscript, you have to give details about that.

Answer: we revised the abbreviations in the paper, including X-STRs and Y-STRs. 

3. I recommend that a native speaker of English review the manuscript to improve word choice, sentence structure, and grammar.

Answer: The paper was revised by a native speaker of English (U.S.A). 

4. The conclusion need to clear and specific, with 3 short conclusion.

Answer: We revised the conclusion according the suggestion of the reviewer.

Reviewer #2: 

1. The language and structure of the article are so unclear that the merit can't be assessed. Most of the sentences are overly long, complicated, as well as ungrammatical in some instances. Throughout the article the language is neither clear, nor concise. Typographical and spelling errors have also been encountered.

Answer: The paper was revised by a native speaker of English (U.S.A). 

2. The article gives relations to three cases that are not included in the study groups described in the methods. Therefore part of the article rather looks as a case report (in this case sample sizes are not large enough to produce robust results).

Answer: 

(1) Three complex close relative kinship cases were analyzed to evaluate application of CPI Cutoff Value. At the same time, in order to explain the problem that high CPI value (CPI≥10000) will lead to false conclusions in a part of cases.

(2) In our present study, we only had 1442 real trio cases, 803 real duo cases and 3 complex close relative cases typed using Goldeneye 25A kit. We will study more cases in the future.

3. Regarding the tables of this manuscript, they are not all placed directly after the paragrapgh where they were first cited (see Table 2).

Answer: we revised it in the results and discussion.

4. The references formatting does not respect the style of the journal (does not list the first six authors followed by et.al.). Some of the title used as references are rather outdated (over 20 years old).

Answer: we revised the references formatting according to the style of the journal. One reference is over 20 years old (NO. 8), because DNA was extracted according to the method.

5. Laboratory protocols were not made fully availabe. Detailed protocols could enhance the reproductibility of the results.

Answer: We added the laboratory protocols in materials and methods (samples).

6. In part the discussions only repeat the results without interpreting them.

Answer: we amended the part of the discussion.

---

## [Editor Report · Decision Letter 1]

31 Oct 2019

Application of CPI Cutoff Value Based on Parentage Testing of Duos and Trios Typed by Four Autosomal Kits

PONE-D-19-24597R1

Dear Dr. Zhang,

We are pleased to inform you that your manuscript has been judged scientifically suitable for publication and will be formally accepted for publication once it complies with all outstanding technical requirements.

With kind regards,

Alexandru Rogobete, Ph.D., M.Sc., Clin Res

Academic Editor

PLOS ONE

Additional Editor Comments (optional):

Dear Authors,

I read your paper carefully and it looks much better.

Thank you

BR,
---

## [Editor Report · Acceptance letter]

5 Nov 2019

PONE-D-19-24597R1 

Application of CPI Cutoff Value Based on Parentage Testing of Duos and Trios Typed by Four Autosomal Kits 

Dear Dr. Zhang:

I am pleased to inform you that your manuscript has been deemed suitable for publication in PLOS ONE. Congratulations! Your manuscript is now with our production department. 

With kind regards,

on behalf of

Dr. Alexandru Rogobete 

Academic Editor

PLOS ONE